# Deep Residual Learning-Based Classification with Identification of Incorrect Predictions and Quantification of Cellularity and Nuclear Morphological Features in Digital Pathological Images of Common Astrocytic Tumors

**DOI:** 10.3390/cancers16132449

**Published:** 2024-07-03

**Authors:** Yen-Chang Chen, Shinn-Zong Lin, Jia-Ru Wu, Wei-Hsiang Yu, Horng-Jyh Harn, Wen-Chiuan Tsai, Ching-Ann Liu, Ken-Leiang Kuo, Chao-Yuan Yeh, Sheng-Tzung Tsai

**Affiliations:** 1Division of Digital Pathology, Department of Anatomical Pathology, Hualien Tzu Chi Hospital, Buddhist Tzu Chi Medical Foundation, Hualien 970, Taiwan; s92312129@gmail.com; 2Department of Pathology, School of Medicine, Tzu Chi University, Hualien 970, Taiwan; 3Institute of Medical Sciences, Tzu Chi University, Hualien 970, Taiwan; shinn-zong@tzuchi.com.tw; 4Bioinnovation Center, Buddhist Tzu Chi Medical Foundation, Hualien 970, Taiwan; arthewduke@gmail.com (H.-J.H.); anita8412@tzuchi.com.tw (C.-A.L.); 5Department of Neuroscience Center, Hualien Tzu Chi Hospital, Buddhist Tzu Chi Medical Foundation, Hualien 970, Taiwan; 6Department of Neurosurgery, Hualien Tzu Chi Hospital, Buddhist Tzu Chi Medical Foundation, Hualien 970, Taiwan; 7Department of Surgery, School of Medicine, Tzu Chi University, Hualien 970, Taiwan; 8Integration Center of Traditional Chinese and Modern Medicine, Hualien Tzu Chi Hospital, Buddhist Tzu Chi Medical Foundation, Hualien 970, Taiwan; u8931246@yahoo.com.tw; 9Department of Medical Research, Hualien Tzu Chi Hospital, Buddhist Tzu Chi Medical Foundation, Hualien 970, Taiwan; 10aetherAI, Co., Ltd., Taipei 115, Taiwan; seanyu@aetherai.com; 11Division of Molecular Pathology, Department of Anatomical Pathology, Hualien Tzu Chi Hospital, Buddhist Tzu Chi Medical Foundation, Hualien 970, Taiwan; 12Department of Pathology, Tri-Service General Hospital, National Defense Medical Center, Taipei 114, Taiwan; doc31779@mail.ndmctsgh.edu.tw; 13YuanLi Instrument Co., Ltd., Taipei 114, Taiwan; derek_yi@yuanli.tw

**Keywords:** digital pathological images, diffuse astrocytoma, anaplastic astrocytoma, glioblastoma, deep residual learning, residual neural network, hybrid task cascade, quantification, cellularity, nuclear morphological feature

## Abstract

**Simple Summary:**

This study presented an artificial intelligence-based classification emphasizing error identification and quantification of cellularity and nuclear morphological features in digital pathological images of common astrocytic tumors. The identification of incorrect predictions was essential for the subsequent development of better techniques. Quantifying cellularity and nuclear morphological features brought deeper insights into neoplastic morphology and paved the way for further development of a scoring system for objective classification and precision diagnosis to improve interobserver variations.

**Abstract:**

Interobserver variations in the pathology of common astrocytic tumors impact diagnosis and subsequent treatment decisions. This study leveraged a residual neural network-50 (ResNet-50) in digital pathological images of diffuse astrocytoma, anaplastic astrocytoma, and glioblastoma to recognize characteristic pathological features and perform classification at the patch and case levels with identification of incorrect predictions. In addition, cellularity and nuclear morphological features, including axis ratio, circularity, entropy, area, irregularity, and perimeter, were quantified via a hybrid task cascade (HTC) framework and compared between different characteristic pathological features with importance weighting. A total of 95 cases, including 15 cases of diffuse astrocytoma, 11 cases of anaplastic astrocytoma, and 69 cases of glioblastoma, were collected in Taiwan Hualien Tzu Chi Hospital from January 2000 to December 2021. The results revealed that an optimized ResNet-50 model could recognize characteristic pathological features at the patch level and assist in diagnosis at the case level with accuracies of 0.916 and 0.846, respectively. Incorrect predictions were mainly due to indistinguishable morphologic overlap between anaplastic astrocytoma and glioblastoma tumor cell area, zones of scant vascular lumen with compact endothelial cells in the glioblastoma microvascular proliferation area mimicking the glioblastoma tumor cell area, and certain regions in diffuse astrocytoma with too low cellularity being misrecognized as the glioblastoma necrosis area. Significant differences were observed in cellularity and each nuclear morphological feature among different characteristic pathological features. Furthermore, using the extreme gradient boosting (XGBoost) algorithm, we found that entropy was the most important feature for classification, followed by cellularity, area, circularity, axis ratio, perimeter, and irregularity. Identifying incorrect predictions provided valuable feedback to machine learning design to further enhance accuracy and reduce errors in classification. Moreover, quantifying cellularity and nuclear morphological features with importance weighting provided the basis for developing an innovative scoring system to achieve objective classification and precision diagnosis among common astrocytic tumors.

## 1. Introduction

The astrocytic tumor is the major type of glioma, which includes glioblastoma, anaplastic astrocytoma, diffuse astrocytoma, and other rare subtypes [1,2,3,4]. Each subtype is characterized by specific pathological features and is assigned to one of four grades (grades I to IV) according to the World Health Organization (WHO) classification based on tumor behavior and prognosis [3,4]. Glioblastoma is categorized as grade IV, anaplastic astrocytoma as grade III, and diffuse astrocytoma as grade II [3,4]. Currently, pathologists perform classification subjectively, but the task remains challenging due to interobserver variations [5,6]. Interobserver variations can affect the accuracy of diagnosis and subsequent treatment decisions [6]. Recently, the superior performance of artificial intelligence (AI)-assisted approaches in digital pathological images has been demonstrated, including target detection, counting, grading, and classification of various diseases [7,8,9,10,11,12,13]. Further, cellularity and nuclear morphological features extracted by the instance segmentation technique enable the classification of diseases and the discovery of prognostic factors [14,15,16,17,18].

Nowadays, the deep residual learning algorithm is the main strategy for image recognition [19]. It is an advanced machine learning technique with a residual neural network (ResNet) to address the degradation problem in conventional deep neural networks [19]. Some research applying different AI models, including deep learning algorithms, in glioma classification has showcased various performances [20,21,22,23]. Moreover, incorrect predictions by AI models have always occurred and have yet to be presented and explained in previous works [20,21,22,23]. Several studies integrating cellularity and nuclear morphologic features into deep learning models have enhanced glioma classification [17,18]. However, the quantification or importance weight of the parameters has yet to be presented [17,18].

In the present study, we leveraged a ResNet [19] in digital pathological images of common astrocytic tumors, specifically the diffuse astrocytoma, anaplastic astrocytoma, and glioblastoma, to recognize characteristic pathological features and perform classification at the patch and case levels. The focus is on the identification and presentation of prediction errors. In addition, we quantified cellularity and nuclear morphological features, including axis ratio, circularity, entropy, area, irregularity, and perimeter, via a hybrid task cascade (HTC) framework. Comparisons between different characteristic pathological features and importance weighting were also performed. The identification of prediction errors has the potential to enhance the machine learning algorithm’s performance. Furthermore, measuring cellularity and nuclear morphological features with weighting could aid in developing a prediction model for common astrocytic tumors.

## 2. Materials and Methods

### 2.1. Cases

Cases of diffuse astrocytoma, anaplastic astrocytoma, and glioblastoma in Taiwan Hualien Tzu Chi Hospital were collected from January 2000 to December 2021. Two pathology doctors made and confirmed their pathological diagnoses in routine medical practice. Each case’s representative pathological hematoxylin-and-eosin-stained (H&E stain) slide was scanned and transformed into digital pathological images. Then, we annotated the regions of interest (ROIs) for characteristic pathological features. The annotated ROIs included diffuse astrocytoma, anaplastic astrocytoma, glioblastoma tumor cell area, glioblastoma microvascular proliferation area, and glioblastoma necrosis area (Figure 1). The study was approved by the Institutional Review Board of Hualien Tzu Chi Hospital (approval No. IRB112-255-B).

### 2.2. Application of Deep Residual Learning Model for Classification

To prepare training data for the development of the classification model, we initially allocated 40% of the digital pathological images to an independent testing set through random selection. The remaining 60% were subdivided into training and validation sets in a 7:3 ratio via random splitting. For each ROI on a slide, we extracted 512 × 512 patches using a stride of 256. Given the irregularity of the ROIs, we excluded patches in the margin that had less than a 50% intersection area with an ROI. Then, we optimized a residual neural network-50 (ResNet-50) [19], a convolutional neural network with a depth of 50 layers and a residual learning algorithm, with the prepared dataset. The flowchart of this work is presented in Figure 2. The results are presented using a confusion matrix, in which the x-axis represents precision and the y-axis represents recall, at the patch and case levels, respectively.

At the patch level, a bootstrapping strategy [24] was adopted, involving 50 samplings of the independent testing set to determine the 95% confidence interval (CI) of prediction results. Two confusion matrices were planned to be presented. One presented all five annotated characteristic pathological features. Another matrix, with the expectation of a partially indistinguishable morphological spectrum between anaplastic astrocytoma and glioblastoma tumor cell area [4], presented four categories in which the anaplastic astrocytoma and glioblastoma tumor cell area were combined into a single category, the high-grade astrocytoma tumor cell area.

At the case level, different inclusion criteria for the patch ratio of the predicted characteristic pathological feature to be considered for classification were tested to filter out minor errors. The criteria we used were 0.00, 0.02, and 0.05. The inclusion criterion of 0.00 meant that any predicted characteristic pathological feature patch ratio was considered for classification. The inclusion criteria of 0.02 and 0.05 meant that the patch ratios of the predicted characteristic pathological features could only be considered for classification if they reached 0.02 and 0.05, respectively.

### 2.3. Quantification of Cellularity and Nuclear Morphological Features with Importance Weighting

For nucleus detection, we applied a hybrid task cascade (HTC) framework for instance segmentation [14]. After that, cellularity and nuclear morphological features, including axis ratio, circularity, entropy, area, irregularity, and perimeter, were evaluated. Cellularity was the ratio of the area occupied by cells to the total area (Figure 3). Welch’s *t*-test with Bonferroni correction [25] was used to test whether the means of cellularity were equal for all two-group combinations of the characteristic pathological features. A *p*-value less than 0.05 is considered statistically significant. For nuclear morphological features, we measured the lengths of both the long and short axes in each nucleus and calculated the long-to-short axis ratio. Circularity was determined by the pixel overlap between a concentric circle and the nucleus. The entropy quantified pixel randomness. The area denoted nuclear size in square micrometers. Irregularity represented the length variance from the nucleus center to each boundary vertex. Perimeter was the estimated total length along a nuclear boundary in micrometers. The nuclear morphological features are sketched in Figure 4. For each nuclear morphological feature, four commonly used statistical moments (mean, variance, skewness, and kurtosis) were computed for comparisons [26,27]. In addition, one-way analysis of variance (ANOVA) [28] was used to test the differences in each moment of each nuclear morphological feature between the characteristic pathological features. A *p*-value less than 0.05 is considered statistically significant. A decision-tree-based machine learning algorithm, extreme gradient boosting (XGBoost) [29], was applied to evaluate the importance weights. A bootstrapping strategy [24] was adopted, involving 50 samplings of the independent testing set to determine the 95% confidence interval (CI) of importance weights.

## 3. Results

### 3.1. Data Summary

A total of 95 cases were collected, including 15 cases of diffuse astrocytoma, 11 cases of anaplastic astrocytoma, and 69 cases of glioblastoma. Among these cases, 39, 17, and 39 were randomly split into training, validation, and testing sets. The sum of annotated ROIs was 10,920, producing 61,696 patches of 512 × 512 pixels. The details of cases, ROIs, and patches are presented in Table 1.

### 3.2. Outcomes of Application of Deep Residual Learning Model for Classification

#### 3.2.1. At the Patch Level

Using ResNet-50 to recognize the characteristic pathological features, the accuracy was 0.916 (95% CI, 0.915–0.916) at the patch level. The details of the results are shown in Figure 5, in which most patches fell on the diagonal line from upper left to lower right, representing correct predictions. However, two principal incorrect predictions occurred (marked by gray background lattices in Figure 5). Around 67.9% of patches from anaplastic astrocytoma were misrecognized as glioblastoma tumor cell area. In addition, around 38.5% of patches from the glioblastoma microvascular proliferation area were also misrecognized as glioblastoma tumor cell area.

An accuracy of 0.960 (95% CI, 0.960–0.961) was achieved at the patch level after combining anaplastic astrocytoma and glioblastoma tumor cell area into a single category of high-grade astrocytoma tumor cell area (Figure 6). In the confusion matrix, the majority of patches were correctly predicted. Nevertheless, a main incorrect prediction was presented (lattice with gray background in Figure 6). Approximately 34.7% of patches from the glioblastoma microvascular proliferation area were erroneously categorized into high-grade astrocytoma tumor cell area.

#### 3.2.2. At the Case Level

Based on inclusion criterion of 0.00, the accuracy attained using ResNet-50 for classifying diffuse astrocytoma, anaplastic astrocytoma, and glioblastoma was 0.769 at the case level. The details of the results are shown in Figure 7, in which six cases of diffuse astrocytoma and three cases of anaplastic astrocytoma were incorrectly classified as cases of glioblastoma.

Based on the inclusion criterion of 0.02, the accuracy obtained using ResNet-50 for classifying diffuse astrocytoma, anaplastic astrocytoma, and glioblastoma at the case level was 0.846 (Figure 8). Three cases of diffuse astrocytoma and three cases of anaplastic astrocytoma were incorrectly classified as cases of glioblastoma. Based on the inclusion criterion of 0.05, the accuracy was 0.846 at the case level (Figure 9). Two cases of diffuse astrocytoma and three cases of anaplastic astrocytoma were incorrectly classified as cases of glioblastoma. A case of diffuse astrocytoma was incorrectly classified as anaplastic astrocytoma.

The details of testing results using ResNet-50 for classifying diffuse astrocytoma, anaplastic astrocytoma, and glioblastoma are shown in Appendix A of the Appendix A. Table 2 presents the selective cases with incorrect classification from Appendix A. Among these, three cases of diffuse astrocytoma (cases 13, 17, and 22) were incorrectly classified into glioblastoma due to the presence of erroneously predicted scant patches of glioblastoma necrosis area or glioblastoma microvascular proliferation area based on the inclusion criterion of 0.00. Moreover, one case of diffuse astrocytoma (case 14) was wrongly classified as glioblastoma, attributed to the small number of erroneous prediction patches in the glioblastoma tumor cell area, glioblastoma necrosis area, and glioblastoma microvascular proliferation area. Additionally, another diffuse astrocytoma case (case 25) was incorrectly classified as glioblastoma based on the inclusion criteria of 0.00 and 0.02, stemming from the presence of falsely predicted minor patches of glioblastoma tumor cell area, glioblastoma necrosis area, and glioblastoma microvascular proliferation area. This case was also misclassified as anaplastic astrocytoma with the criterion of 0.05 due to the incorrectly predicted major patches of anaplastic astrocytoma. Furthermore, one instance of diffuse astrocytoma (case 38) was inaccurately classified as glioblastoma because of the falsely predicted major patches of glioblastoma necrosis area. Lastly, three cases of anaplastic astrocytoma (cases 15, 27, and 29) were misclassified as glioblastoma due to the significant number of incorrect prediction patches of glioblastoma tumor cell areas.

### 3.3. Outcomes of Quantification of Cellularity and Nuclear Morphological Features with Importance Weighting

#### 3.3.1. Cellularity

The average cellularities for individual characteristic pathological features are shown in Table 3. Glioblastoma tumor cell area exhibited the highest cellularity at 0.195 ± 0.051 (mean ± standard deviation [SD]), followed by anaplastic astrocytoma with a cellularity of 0.180 ± 0.063, and glioblastoma microvascular proliferation area, which had a cellularity of 0.122 ± 0.052. The diffuse astrocytoma displayed a cellularity of 0.052 ± 0.018. The glioblastoma necrosis area exhibited the lowest cellularity, measured at 0.003 ± 0.008. Welch’s *t*-test with Bonferroni correction revealed that the cellularities were significantly different between all two-group combinations (Table 4).

#### 3.3.2. Nuclear Morphological Features

The nuclear morphological features in individual characteristic pathological features are presented in Table 5. The average axis ratio ranged from 1.437 ± 0.103 to 1.734 ± 0.144, circularity from 0.574 ± 0.035 to 0.672 ± 0.037, entropy from 4.711 ± 0.227 to 4.925 ± 0.202, area from 13.676 ± 3.536 to 30.546 ± 6.183 μm, irregularity from 2.648 ± 1.115 to 7.784 ± 2.161, and perimeter from 14.327 ± 1.827 to 21.745 ± 2.075 μm. The one-way ANOVA analysis revealed significant differences in both mean and variance for all six individual nuclear morphological features among the various categories. Furthermore, there were statistically significant variations in the skewness and kurtosis of the axis ratio, circularity, and irregularity among the different categories. However, the skewness and kurtosis values for entropy, area, and perimeter were not statistically significant in discriminating between these categories.

#### 3.3.3. Importance Weighting

By using XGBoost to evaluate importance weight, entropy was the most important with a weight of 1607.2 (95% CI, 1600.2–1614.2), followed by cellularity with an important weight of 1160.4 (95% CI, 1153.9–1166.9), area, 528.8 (95% CI, 524.4–533.2), circularity, 524.2 (95% CI, 519.7–528.7), axis ratio, 502.1 (95% CI, 498.6–505.6), perimeter, 434.8 (95% CI, 432.3–437.3), and irregularity with an importance weight of 358.0 (95% CI, 355.9–360.1) (Figure 10).

## 4. Discussion

This study presented a deep residual learning-based classification focusing on identifying incorrect predictions and quantification of cellularity and nuclear morphological features with importance weighting specifically in common astrocytic tumors.

A patch size of 512 × 512 pixels was chosen to balance graphics processing unit (GPU) video random access memory (VRAM) limitations and the need to consider a more extensive region within each patch. While larger patches could provide more information for the model, they also require more GPU VRAM. In our case, a 512 × 512 patch size requires approximately 20 GB of VRAM, which fits within the 24 GB VRAM capacity of our display card. Additionally, using a 512 × 512 patch size is common in previous AI-related studies [30,31,32]. For the stride length, we aimed to balance storage requirements and the desire for the model to see all possible regions. An extreme approach would be to set the stride to one pixel, ensuring that all possible patches are provided. However, this would result in excessive patches, exceeding our server’s storage capacity. Setting the stride to half the patch size (256 pixels) balances storage requirements and region coverage well.

We demonstrated the excellent performance of the ResNet-50 for recognizing characteristic pathologic features and classification in common astrocytic tumors, compared to other similar previous works [20,21,22,23]. However, what distinguished our work from others was that we identified and explained the details of incorrect predictions, which were not articulated in previous works. Our study revealed three main incorrect predictions (Figure 11). Firstly, partial indistinguishable morphological overlap led to anaplastic astrocytoma being misrecognized as glioblastoma tumor cell area at the patch level and classified as glioblastoma at the case level. Secondly, certain zones within the glioblastoma microvascular proliferation area presented dense and crowded endothelial cells with scarce vascular lumen mimicking the glioblastoma tumor cell area. Thirdly, the cellularity in certain regions of diffuse astrocytoma was too low, leading to misrecognition as the glioblastoma necrosis area at the patch level and glioblastoma at the case level. As suggested by our results, the selection of an inclusion criterion of 0.02 for the patch ratio considered for classification was appropriate to filter out minor errors and reduce misclassification.

Several studies have applied a deep learning model integrating cellularity and nuclear morphologic features to enhance glioma classification [17,18]. However, no quantified value or importance weight is available. Our study quantified cellularity and nuclear morphological features, including axis ratio, circularity, entropy, area, irregularity, and perimeter, with an evaluation of importance weight. In addition, the comparisons of the above attributes showed significant differences among different characteristic pathological features. Furthermore, using the XGBoost algorithm, we found that entropy was the most important feature for classification. The entropy of an image indicates the level of randomness present [21,33]. Previous studies demonstrated the value of entropy in low-grade gliomas was different than that of high-grade gliomas [21], and the entropy could differentiate patients with glioblastoma from a healthy control in pathological images [33]. The entropy of the nucleus reflects spatial homogeneity/heterogeneity [21], which is potentially influenced by factors like the extent of nuclear anaplasia and pleomorphism in neoplastic cells, suggesting the potential reason for this importance. Further large-scale studies applying advanced statistical methods, such as a receiver operating characteristic (ROC) curve [34,35], logistic regression model [36], or nomogram [37], to determine the cut-off value of each attribute are needed to establish a prediction model for differentiation among common astrocytic tumors.

Our study had several limitations. Firstly, the case number was relatively small, potentially requiring a more robust dataset that might yield more powerful and convincing results. Secondly, this study only included cases from a single hospital. Including a more diverse and representative selection of cases could enhance the ability to extract disease characteristics more effectively. Thirdly, the study did not include other rare subtypes of astrocytic tumors that were difficult to collect, suggesting a potential area for further exploration and inclusion to provide a more comprehensive understanding of the astrocytic tumors.

## 5. Conclusions

This study demonstrated a deep residual learning-based classification focusing on identifying and explaining the incorrect predictions and quantification of cellularity and nuclear morphological features with importance weighting specifically in common astrocytic tumors. Identifying incorrect predictions provided worthwhile feedback to machine learning algorithm design to further refine accuracy and diminish errors in classification. Quantifying cellularity and nuclear morphological features with importance weighting was the basis for developing an innovative scoring system. Further multicentric study involving more significant case numbers and other rare subtypes of astrocytic tumors is necessary to establish a powerful prediction model to achieve objective classification and precision diagnosis among astrocytic tumors.

## Figures and Tables

**Figure 1 cancers-16-02449-f001:**
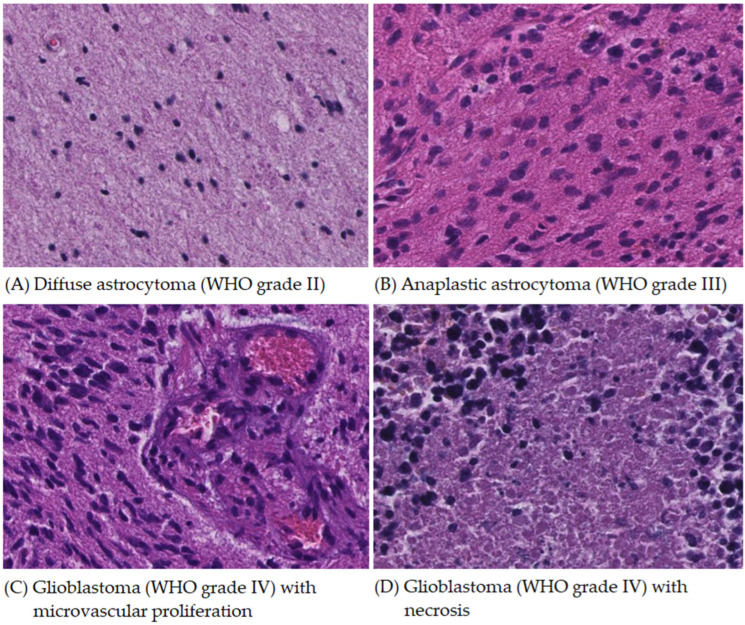
Representative images and the annotated regions of interest (ROIs) included the following: (**A**) diffuse astrocytoma showing nuclear atypia of neoplastic astrocytes, (**B**) anaplastic astrocytoma in which neoplastic astrocytes show nuclear atypia with anaplasia, (**C**) glioblastoma tumor cell area (**left**), composed of neoplastic astrocytes with nuclear atypia and anaplasia, and microvascular proliferation (**right**) with vascular lumen filled with blood cells and lined by endothelial cells, and (**D**) glioblastoma necrosis area presenting the contours of dead cells mixed with a small number of viable cells. The cellularity is moderately increased in diffuse astrocytoma, while high cellularity is usually presented in anaplastic astrocytoma and glioblastoma. (H&E stain, all pictures in the same magnification of 200×).

**Figure 2 cancers-16-02449-f002:**
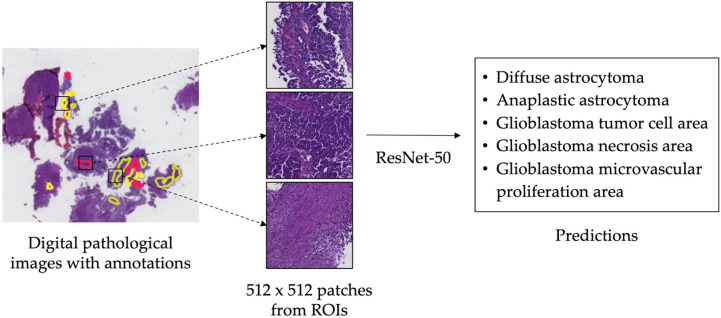
The flowchart for applying residual neural network-50 (ResNet-50) to predict characteristic pathological features. Regions of interest (ROIs) were annotated on digital pathological images, and 512 × 512 patches were extracted from ROIs by using a stride of 256. Residual neural network-50 (ResNet-50) was utilized to generate predictions.

**Figure 3 cancers-16-02449-f003:**
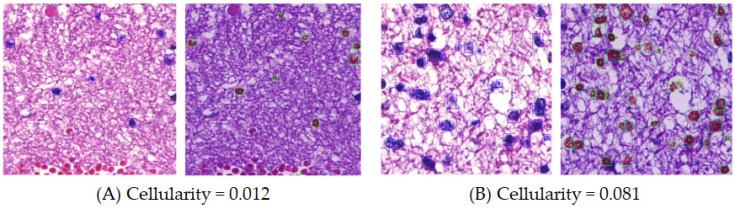
Examples of cellularity. Cellularity represents the proportion of cell-occupied area to the overall area. The green boxes indicate the cells detected. (**A**) Cellularity of 0.012 and (**B**) cellularity of 0.081.

**Figure 4 cancers-16-02449-f004:**
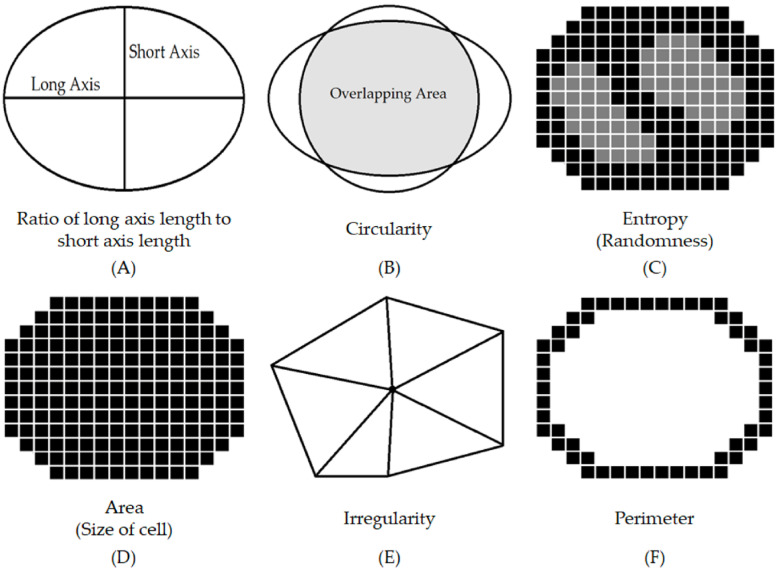
The nuclear morphological features. The attributes we used in the present study and the definitions are as follows: (**A**) axis ratio, the ratio of the length of the long axis to the short axis; (**B**) circularity, the overlapping area between the nucleus and a concentric circle; (**C**) entropy, the measurement of pixel randomness; (**D**) area, the estimated nuclear size in square micrometers; (**E**) irregularity, the length variance from the center of a nucleus to each vertex of nuclear boundary; and (**F**) perimeter, the total length along a nuclear contour in micrometers.

**Figure 5 cancers-16-02449-f005:**
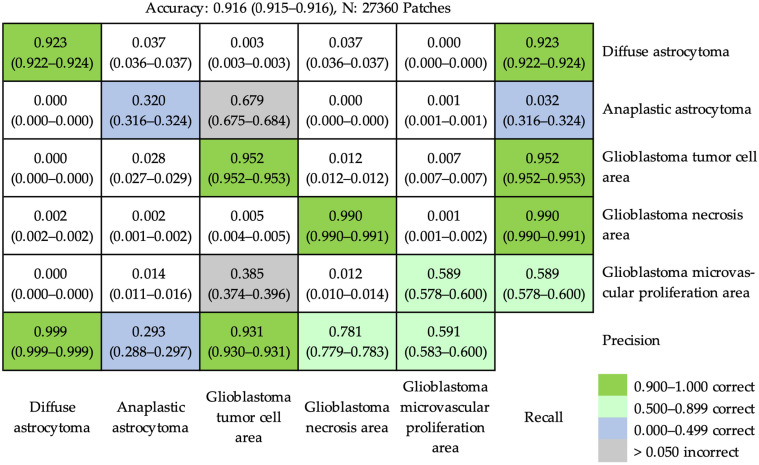
The confusion matrix of prediction results using ResNet-50 to recognize characteristic pathological features at the patch level. The accuracy was 0.916 (95% confidence interval, 0.915–0.916). Most patches were correctly predicted. Two main incorrect predictions (lattices with gray background) were that partial patches (67.9%) of anaplastic astrocytoma and some patches (38.5%) of glioblastoma microvascular proliferation area were misrecognized as glioblastoma tumor cell area.

**Figure 6 cancers-16-02449-f006:**
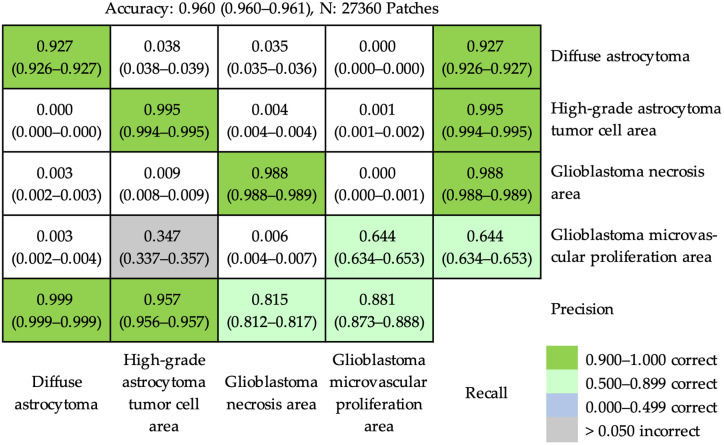
The confusion matrix of prediction results using ResNet-50 to recognize diffuse astrocytoma, high-grade astrocytoma tumor cell area, glioblastoma necrosis area, and glioblastoma microvascular proliferation area at the patch level. The accuracy achieved was 0.960 (95% confidence interval, 0.960–0.961). Most patches were correctly predicted. A principal incorrect prediction (lattice marked by gray background) was that some patches (34.7%) of the glioblastoma microvascular proliferation area were erroneously categorized into high-grade astrocytoma tumor cell area.

**Figure 7 cancers-16-02449-f007:**
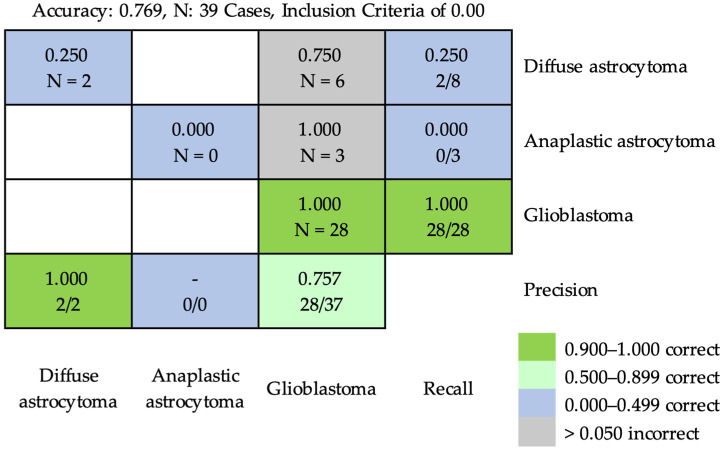
The confusion matrix of prediction results using ResNet-50 for classifying diffuse astrocytoma, anaplastic astrocytoma, and glioblastoma at the case level, based on inclusion criterion of 0.00. The inclusion criterion of 0.00 meant that any predicted characteristic pathological feature patch ratio was considered for classification. The accuracy was 0.769. Six cases of diffuse astrocytoma and three cases of anaplastic astrocytoma were incorrectly classified into cases of glioblastoma.

**Figure 8 cancers-16-02449-f008:**
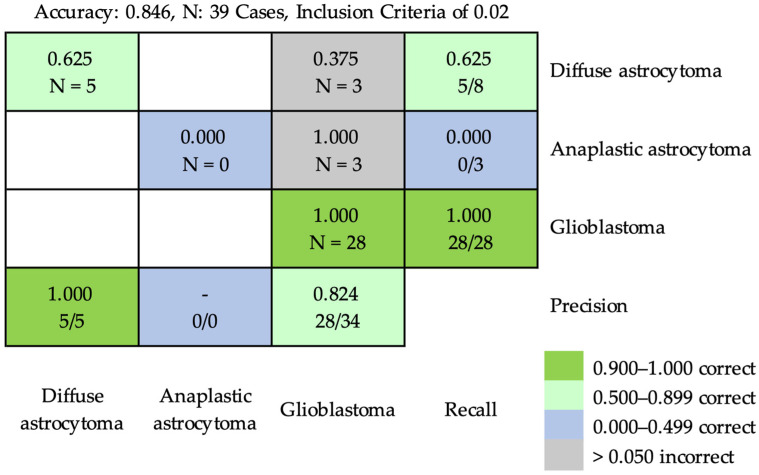
The confusion matrix of prediction results using ResNet-50 for classifying diffuse astrocytoma, anaplastic astrocytoma, and glioblastoma at the case level, based on the inclusion criterion of 0.02. The inclusion criterion of 0.02 meant that the patch ratios of the predicted characteristic pathological features could only be considered for classification if they reached 0.02. The accuracy was 0.846. Three cases of diffuse astrocytoma and three cases of anaplastic astrocytoma were incorrectly classified as cases of glioblastoma.

**Figure 9 cancers-16-02449-f009:**
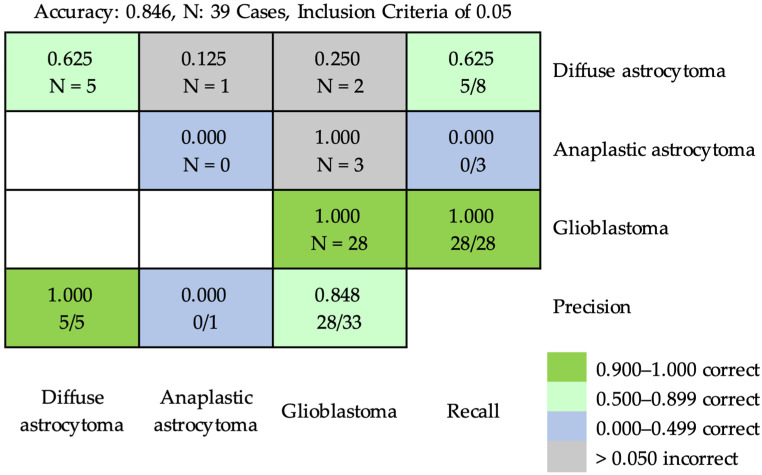
The confusion matrix of prediction results using ResNet-50 for classifying diffuse astrocytoma, anaplastic astrocytoma, and glioblastoma at the case level, based on inclusion criterion of 0.05. Inclusion criterion of 0.05 meant that the patch ratios of the predicted characteristic pathological features could only be considered for classification if they reached 0.05. The accuracy was 0.846. Two cases of diffuse astrocytoma and three cases of anaplastic astrocytoma were incorrectly classified into cases of glioblastoma. A case of diffuse astrocytoma was incorrectly classified as anaplastic astrocytoma.

**Figure 10 cancers-16-02449-f010:**
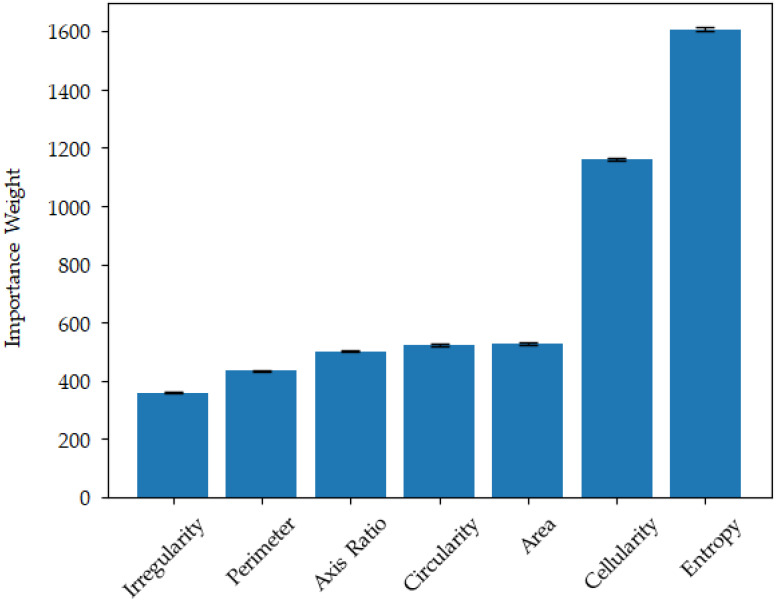
The importance weight evaluated by using the XGBoost model. Entropy was the most significant contributor, followed by cellularity, area, circularity, axis ratio, perimeter, and irregularity.

**Figure 11 cancers-16-02449-f011:**
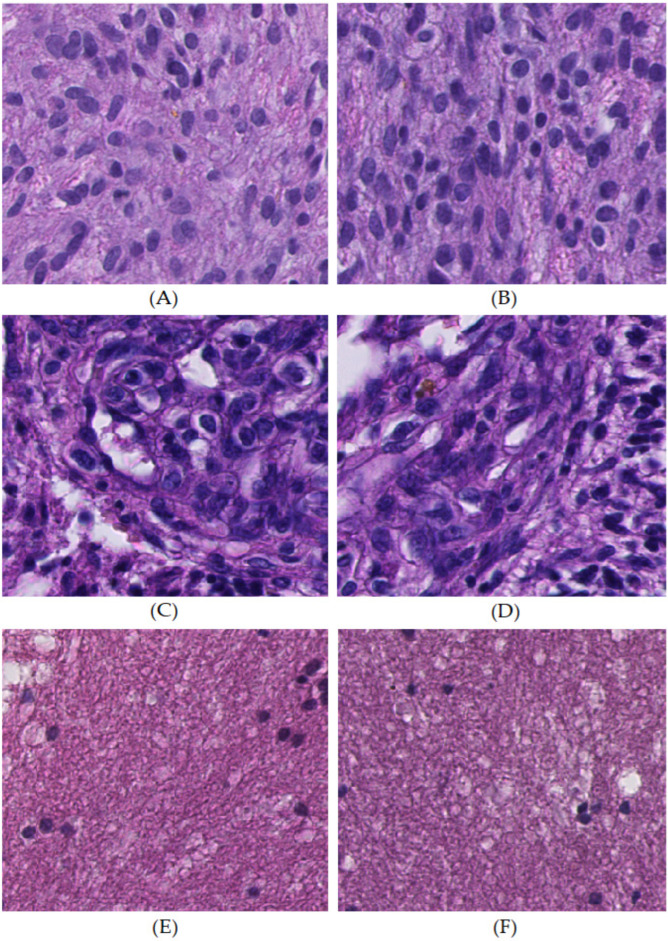
The actual case examples of incorrect predictions. (**A**,**B**) Anaplastic astrocytoma resembled and was misrecognized as a glioblastoma tumor cell area. (**C**,**D**) Densely compact endothelial cells with a small amount of vessel lumen in the glioblastoma microvascular proliferation area mimicked the glioblastoma tumor cell area. (**E**,**F**) Low cellularity regions in diffuse astrocytoma imitated glioblastoma necrosis area.

**Table 1 cancers-16-02449-t001:** Data summary.

Category	Training			Validation			Testing		
	Cases	ROIs	Patches	Cases	ROIs	Patches	Cases	ROIs	Patches
Diffuse astrocytoma	4	729	15,176	3	212	1877	8	833	12,316
Anaplastic astrocytoma	6	1487	3551	2	177	401	3	256	1014
Glioblastoma tumor cell area		2207	7382		515	913		1521	11,653
Glioblastoma necrosis area	29	657	2357	12	1072	2088	28	460	2145
Glioblastoma microvascular proliferation area		423	465		114	126		257	232
Total	39	5503	28,931	17	2090	5405	39	3327	27,360

Abbreviation: ROIs, regions of interest.

**Table 2 cancers-16-02449-t002:** The selective testing cases with incorrect classification by using ResNet-50 for classifying diffuse astrocytoma, anaplastic astrocytoma, and glioblastoma.

Testing Case No.	Diagnosis	Characteristic Morphological Features	Prediction		Inclusion Criterion of 0.00		Inclusion Criterion of 0.02		Inclusion Criterion of 0.05	
			Count	Ratio	Ratio	Classification	Ratio	Classification	Ratio	Classification
Case 13	Diffuse astrocytoma	Diffuse astrocytoma	5025	0.998	0.998	** Glioblastoma **	**0.998**	**Diffuse astrocytoma**	**0.998**	**Diffuse astrocytoma**
		Anaplastic astrocytoma	3	0.001	0.001		0.001		0.001	
		Glioblastoma tumor cell area								
		Glioblastoma necrosis area	9	0.002	**0.002**		0.002		0.002	
		Glioblastoma microvascular proliferation area								
Case 14	Diffuse astrocytoma	Diffuse astrocytoma	216	0.722	0.722	** Glioblastoma **	0.722	** Glioblastoma **	0.722	** Glioblastoma **
		Anaplastic astrocytoma								
		Glioblastoma tumor cell area	22	0.074	**0.074**		**0.074**		**0.074**	
		Glioblastoma necrosis area	60	0.201	**0.201**		**0.201**		**0.201**	
		Glioblastoma microvascular proliferation area	1	0.003	**0.003**		0.003		0.003	
Case 15	Anaplastic astrocytoma	Diffuse astrocytoma				** Glioblastoma **		** Glioblastoma **		** Glioblastoma **
		Anaplastic astrocytoma								
		Glioblastoma tumor cell area	146	1.000	**1.000**		**1.000**		**1.000**	
		Glioblastoma necrosis area								
		Glioblastoma microvascular proliferation area								
Case 17	Diffuse astrocytoma	Diffuse astrocytoma	5280	0.998	0.998	** Glioblastoma **	**0.998**	**Diffuse astrocytoma**	**0.998**	**Diffuse astrocytoma**
		Anaplastic astrocytoma								
		Glioblastoma tumor cell area								
		Glioblastoma necrosis area	9	0.002	**0.002**		0.002		0.002	
		Glioblastoma microvascular proliferation area								
Case 22	Diffuse astrocytoma	Diffuse astrocytoma	530	0.994	0.994	** Glioblastoma **	**0.994**	**Diffuse astrocytoma**	**0.994**	**Diffuse astrocytoma**
		Anaplastic astrocytoma								
		Glioblastoma tumor cell area								
		Glioblastoma necrosis area	1	0.002	**0.002**		0.002		0.002	
		Glioblastoma microvascular proliferation area	2	0.004	**0.004**		0.004		0.004	
Case 25	Diffuse astrocytoma	Diffuse astrocytoma	1	0.002	0.002	** Glioblastoma **	0.002	** Glioblastoma **	0.002	** Anaplastic astrocytoma **
		Anaplastic astrocytoma	449	0.943	0.943		0.943		**0.943**	
		Glioblastoma tumor cell area	16	0.034	**0.034**		**0.034**		0.034	
		Glioblastoma necrosis area	8	0.017	**0.017**		0.017		0.017	
		Glioblastoma microvascular proliferation area	2	0.004	**0.004**		0.004		0.004	
Case 27	Anaplastic astrocytoma	Diffuse astrocytoma				** Glioblastoma **		** Glioblastoma **		** Glioblastoma **
		Anaplastic astrocytoma	28	0.113	0.113		0.113		0.113	
		Glioblastoma tumor cell area	220	0.887	**0.887**		**0.887**		**0.887**	
		Glioblastoma necrosis area								
		Glioblastoma microvascular proliferation area								
Case 29	Anaplastic astrocytoma	Diffuse astrocytoma				** Glioblastoma **		** Glioblastoma **		** Glioblastoma **
		Anaplastic astrocytoma	296	0.477	0.477		0.477		0.477	
		Glioblastoma tumor cell area	323	0.521	**0.521**		**0.521**		**0.521**	
		Glioblastoma necrosis area								
		Glioblastoma microvascular proliferation area	1	0.002	**0.002**		0.002		0.002	
Case 38	Diffuse astrocytoma	Diffuse astrocytoma	6	0.016	0.016	** Glioblastoma **	0.016	** Glioblastoma **	0.016	** Glioblastoma **
		Anaplastic astrocytoma								
		Glioblastoma tumor cell area								
		Glioblastoma necrosis area	367	0.984	**0.984**		**0.984**		**0.984**	
		Glioblastoma microvascular proliferation area								

The inclusion criterion of 0.00 meant any predicted characteristic pathological feature patch ratio was considered for classification. The inclusion criteria of 0.02 and 0.05 meant that the patch ratios of the predicted characteristic pathological features could only be considered for classification if they reached 0.02 and 0.05, respectively. Number with strikethrough means criteria exclude the patch ratio. The bold number in the ratio column indicates the determinant(s) for classification. Red text indicates incorrect classification.

**Table 3 cancers-16-02449-t003:** The average cellularities for individual characteristic pathological features.

Category	ROIs	Cellularity
Diffuse astrocytoma	1774	0.052 ± 0.018
Anaplastic astrocytoma	1915	0.180 ± 0.063
Glioblastoma tumor cell area	4238	0.195 ± 0.051
Glioblastoma necrosis area	2184	0.003 ± 0.008
Glioblastoma microvascular proliferation area	794	0.122 ± 0.052

Abbreviation: ROI, region of interest. Cellularity is presented by mean ± standard deviation.

**Table 4 cancers-16-02449-t004:** The *p*-value of Welch’s *t*-test with Bonferroni correction for cellularities between characteristic pathological features.

	Diffuse Astrocytoma	Anaplastic Astrocytoma	Glioblastoma Tumor Cell Area	Glioblastoma Necrosis Area	Glioblastoma Microvascular Proliferation Area
Diffuse astrocytoma		<0.001 *	<0.001 *	<0.001 *	<0.001 *
Anaplastic astrocytoma			<0.001 *	<0.001 *	<0.001 *
Glioblastoma tumor cell area				<0.001 *	<0.001 *
Glioblastoma necrosis area					<0.001 *
Glioblastoma microvascular proliferation area					

* Statistically significant, *p* < 0.05.

**Table 5 cancers-16-02449-t005:** The nuclear morphological features for individual characteristic pathological features.

Attributes	Moments (Mean ± SD)	Diffuse Astrocytoma	Anaplastic Astrocytoma	Glioblastoma Tumor Cell Area	Glioblastoma Necrosis Area	Glioblastoma Microvascular Proliferation Area	*F*-Test	
		N = 15	N = 11	N = 69			F-Statistics	*p* Value
Axis Ratio	Mean	1.437 ± 0.103	1.540 ± 0.118	1.570 ± 0.123	1.504 ± 0.193	1.734 ± 0.144	20.377	<0.001 *
	Variance	0.139 ± 0.066	0.192 ± 0.092	0.194 ± 0.101	0.169 ± 0.195	0.359 ± 0.182	12.489	<0.001 *
	Skewness	2.351 ± 0.285	2.113 ± 0.409	1.844 ± 0.332	1.704 ± 0.903	1.839 ± 0.614	10.38	<0.001 *
	Kurtosis	10.106 ± 2.979	9.264 ± 4.545	6.201 ± 2.959	5.264 ± 6.070	5.368 ± 3.936	21.733	<0.001 *
Circularity	Mean	0.672 ± 0.037	0.634 ± 0.040	0.622 ± 0.038	0.641 ± 0.053	0.574 ± 0.035	25.248	<0.001 *
	Variance	0.014 ± 0.003	0.015 ± 0.003	0.015 ± 0.003	0.013 ± 0.007	0.018 ± 0.003	10.478	<0.001 *
	Skewness	−0.642 ± 0.153	−0.422 ± 0.153	−0.330 ± 0.163	−0.423 ± 0.365	−0.250 ± 0.223	11.425	<0.001 *
	Kurtosis	0.140 ± 0.308	−0.235 ± 0.274	−0.341 ± 0.248	−0.282 ± 0.657	−0.525 ± 0.307	11.417	<0.001 *
Entropy	Mean	4.921 ± 0.198	4.745 ± 0.165	4.808 ± 0.187	4.925 ± 0.202	4.711 ± 0.227	7.833	<0.001 *
	Variance	0.127 ± 0.035	0.131 ± 0.034	0.135 ± 0.035	0.138 ± 0.072	0.192 ± 0.059	9.655	<0.001 *
	Skewness	−0.401 ± 0.187	−0.398 ± 0.229	−0.408 ± 0.290	−0.474 ± 0.453	−0.517 ± 0.348	0.806	0.546
	Kurtosis	0.993 ± 0.568	0.885 ± 0.568	0.900 ± 0.692	0.766 ± 1.745	0.866 ± 0.941	0.323	0.899
Area (μm^2^)	Mean	21.311 ± 3.466	25.958 ± 4.901	30.546 ± 6.183	13.676 ± 3.536	25.345 ± 6.750	53.112	<0.001 *
	Variance	83.678 ± 40.508	164.005 ± 78.404	231.174 ± 126.803	55.408 ± 37.289	185.253 ± 181.007	13.109	<0.001 *
	Skewness	1.460 ± 0.377	1.588 ± 0.355	1.471 ± 0.511	1.575 ± 1.037	1.362 ± 0.567	1.639	0.151
	Kurtosis	6.493 ± 7.097	4.493 ± 2.177	4.257 ± 5.202	6.362 ± 14.965	3.129 ± 3.774	1.153	0.334
Irregularity	Mean	3.339 ± 0.948	5.342 ± 1.327	6.783 ± 1.907	2.648 ± 1.115	7.784 ± 2.161	59.537	<0.001 *
	Variance	17.540 ± 9.779	40.655 ± 15.941	61.880 ± 32.027	9.060 ± 9.498	90.134 ± 115.829	9.488	<0.001 *
	Skewness	4.279 ± 0.728	3.691 ± 0.865	3.200 ± 0.663	2.710 ± 1.650	2.747 ± 1.243	7.622	<0.001 *
	Kurtosis	33.584 ± 11.747	26.882 ± 14.452	18.175 ± 9.300	14.466 ± 17.311	13.132 ± 15.371	9.002	<0.001 *
Perimeter (μm)	Mean	17.887 ± 1.390	19.942 ± 1.760	21.745 ± 2.075	14.327 ± 1.827	20.254 ± 2.39	79.245	<0.001 *
	Variance	14.608 ± 5.064	24.255 ± 5.941	30.381 ± 8.932	13.589 ± 7.075	30.875 ± 13.953	24.395	<0.001 *
	Skewness	0.824 ± 0.205	1.014 ± 0.313	0.833 ± 0.300	0.827 ± 0.509	0.846 ± 0.405	1.413	0.221
	Kurtosis	2.137 ± 1.224	1.732 ± 0.952	1.190 ± 1.089	1.392 ± 3.147	1.062 ± 1.275	1.35	0.245

Abbreviation: SD, standard deviation. * Statistically significant, *p* < 0.05.

## Data Availability

The datasets and codes used and analyzed in this study are available from the corresponding authors upon reasonable request.

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
