# Peer review of "Deep Residual Learning-Based Classification with Identification of Incorrect Predictions and Quantification of Cellularity and Nuclear Morphological Features in Digital Pathological Images of Common Astrocytic Tumors"

_cancers, 2024, doi:10.3390/cancers16132449_

Round 1

Reviewer 1 Report

Comments and Suggestions for Authors

This paper presented a deep residual learning-based classification focusing on incorrect predictions and quantification of cellularity and nuclear morphological features in gliomas.  The study is focused on common astrocytic tumors.

The paper is well written and results are in line with the aim of the study. However the homogeneity of the study (case mix) is not guaranteed (A total of 95 cases, including 15 cases of diffuse astrocytoma, 11 cases of anaplastic astrocytoma, and 69 cases of glioblastoma) and the overall number is poor for getting significative results.

However, for its novelty, the paper is suitable for publication.

Author Response

Comments 1: [This paper presented a deep residual learning-based classification focusing on incorrect predictions and quantification of cellularity and nuclear morphological features in gliomas.  The study is focused on common astrocytic tumors. The paper is well written and results are in line with the aim of the study. However the homogeneity of the study (case mix) is not guaranteed (A total of 95 cases, including 15 cases of diffuse astrocytoma, 11 cases of anaplastic astrocytoma, and 69 cases of glioblastoma) and the overall number is poor for getting significative results. However, for its novelty, the paper is suitable for publication.]

Response 1: Thank you for your comments and suggestions. We know these are our limitations in this study. We hope to plan and perform further multicentric studies including more case numbers to get significative results in the future. Thank you for allowing us to publish this study first.

Reviewer 2 Report

Comments and Suggestions for Authors

Please specify the statistical significance threshold used for the Welch’s t-test and ANOVA. Mention if corrections for multiple comparisons was applied, especially given the number of comparisons made.

In the evaluation of importance weighting using XGBoost, provide confidence intervals for the importance scores to show the reliability of these estimates.

Improve the clarity and readability of Figure 3 and Figure 4. The contrast between different elements in the figures can be enhanced to make them more distinguish. Consider use a consistent color scheme and ensure the text is legible.

Standardize the terminology used for referring to the pathological features and categories. Ensure terms such as “anaplastic astrocytoma” and “glioblastoma tumor cell area” are consistently used throughout the paper.

Review and correct typographical errors, such as “celluarity” to “cellularity” and ensure consistent use of medical terms as per standard nomenclature.

Clarify the rationale behind the selection of the patch size (512 x 512 pixels) and stride (256) for the image analysis. Justify why these specific values were choosen and discuss if different values were tested.

Validate the conclusion that entropy is the most important feature for classification. Provide a discussion on how this finding aligns or contradicts existing literature, and suggest potential reasons for this importance.

Author Response

Comment 1: Please specify the statistical significance threshold used for the Welch’s t-test and ANOVA. Mention if corrections for multiple comparisons was applied, especially given the number of comparisons made.

Response 1: Thank you for your comments and suggestions. We used Welch’s t-test with Bonferroni correction to address the issue of multiple comparisons. The statistical significance threshold used for the Welch’s t-test and ANOVA is a p-value less than 0.05. We have added them to the materials and methods (page 5, red texts) and results (page 11, red texts) of the revised manuscript. Thank you for pointing out our miss.

Comment 2: In the evaluation of importance weighting using XGBoost, provide confidence intervals for the importance scores to show the reliability of these estimates.

Response 2: Thank you for your comments and suggestions. We used the bootstrapping strategy involving 50 samplings of the independent testing set to determine the 95% confidence interval (CI) of importance weights. We have added it to the revised manuscript's materials and methods (page 5, red texts). The results with 95% CI were added/revised in the text (pages 11~12, red text), and the result figure (Figure 10) was also revised (page 13). Thank you for your advice.

Comment 3: Improve the clarity and readability of Figure 3 and Figure 4. The contrast between different elements in the figures can be enhanced to make them more distinguish. Consider use a consistent color scheme and ensure the text is legible.

Response 3: Thank you for your comments and suggestions. We have enhanced the contrast between different elements in Figure 3 (page 5) to make them more distinguishable. In addition, we revised and used a consistent color scheme for Figure 4 and ensured the text was legible (page 5). Please see the revised manuscript.

Comment 4: Standardize the terminology used for referring to the pathological features and categories. Ensure terms such as “anaplastic astrocytoma” and “glioblastoma tumor cell area” are consistently used throughout the paper.

Response 4: Thank you for your comments and suggestions. We checked the terminology and ensured terms were consistently used throughout the paper. The “Glioblastoma tumor area” texts in Table 2 (page 10) and Table S1 (supplementary materials) were revised to “Glioblastoma tumor cell area” for consistency. Please remind us if there are any omissions in the manuscript or something we didn't notice.

Comment 5: Review and correct typographical errors, such as “celluarity” to “cellularity” and ensure consistent use of medical terms as per standard nomenclature.

Response 5: Thank you for your comments and suggestions. We reviewed and checked typographical errors and ensured consistent use of medical terms as per standard nomenclature. Please remind us if there are any omissions in the manuscript or something we should have noticed.

Comment 6: Clarify the rationale behind the selection of the patch size (512 x 512 pixels) and stride (256) for the image analysis. Justify why these specific values were chosen and discuss if different values were tested.

Response 6: Thank you for your comments and suggestions. The rationale behind the selection of the patch size (512 x 512 pixels) and stride (256) for the image analysis for the image analysis are following. We also added them in the discussion section of the revised manuscript (page 13, red text).

The patch size of 512 x 512 pixels was chosen to balance graphics processing unit (GPU) video random access memory (VRAM) limitations and the need to consider a more extensive region within each patch. While larger patches could provide more information for the model, they also require more GPU VRAM. In our case, a 512 x 512 patch size requires approximately 20 GB of VRAM, which fits within the 24 GB VRAM capacity of our display card. Additionally, using a 512 x 512 patch size is common in previous AI-related studies [30–32]. For the stride length, we aimed to balance storage requirements and the desire for the model to see all possible regions. An extreme approach would be to set the stride to one pixel, ensuring that all possible patches are provided. However, this would result in excessive patches, exceeding our server's storage capacity. Setting the stride to half the patch size (256 pixels) balances storage requirements and region coverage well.

Comment 7: Validate the conclusion that entropy is the most important feature for classification. Provide a discussion on how this finding aligns or contradicts existing literature, and suggest potential reasons for this importance.

Response 7: Thank you for your comments and suggestions. We reviewed the literature and found two related papers. We added the texts in the discussion of the revised manuscript (page 14, red text) as follows. Thank you for guiding us.

Furthermore, using the XGBoost algorithm, we found that entropy was the most important feature for classification. The entropy of an image indicates the level of randomness present [21,33]. Previous studies demonstrated the value of entropy in low-grade gliomas was different than that of high-grade gliomas [21], and the entropy could differentiate patients with glioblastoma from healthy control in pathological images [33]. The entropy of the nucleus reflects spatial homogeneity/heterogeneity [21], which is potentially influenced by factors like the extent of nuclear anaplasia and pleomorphism in neoplastic cells, suggesting the potential reason for this importance.

Round 2

Reviewer 2 Report

Comments and Suggestions for Authors

After reviewing the revised manuscript, I am pleased with the significant improvements made. The authors have effectively addressed the previous concerns, enhancing the overall quality and ensuring it meets publication standards. I fully support its publication and look forward to its contribution to our field.